# High Poly(ADP-Ribose) Polymerase Expression Does Relate to Poor Survival in Solid Cancers: A Systematic Review and Meta-Analysis

**DOI:** 10.3390/cancers13225594

**Published:** 2021-11-09

**Authors:** Nishant Thakur, Kwangil Yim, Jamshid Abdul-Ghafar, Kyung Jin Seo, Yosep Chong

**Affiliations:** Department of Hospital Pathology, College of Medicine, The Catholic University of Korea, Seoul 07345, Korea; nishantbiotech2014@gmail.com (N.T.); kangse_manse@catholic.ac.kr (K.Y.); jamshid@catholic.ac.kr (J.A.-G.); ywacko@catholic.ac.kr (K.J.S.)

**Keywords:** poly(ADP-ribose) polymerases, neoplasm, BRCA1 and BRCA2, prognosis, meta-analysis

## Abstract

**Simple Summary:**

Poly (ADP-ribose) polymerases (PARPs) are DNA damage repair proteins that are involved in various biological activities ranges from cell proliferation to cell death. The prognostic significance of PARPs is not fully clarified in various cancers. This systematic review aims to reveal the prognostic value of PARP expression in solid cancers and to further correlate with clinicopathological and immunohistochemical markers. Lastly, the inhibition of this pathway through its specific inhibitors may increase the survival of patients with high PARP expression.

**Abstract:**

Poly (ADP-ribose) polymerase (PARP) is a DNA damage repair protein, and its inhibitors have shown promising results in clinical trials. The prognostic significance of PARP is inconsistent in studies of various cancers. In the present study, we conducted a systematic review and meta-analysis to reveal the prognostic and clinicopathological significance of PARP expression in multiple solid cancers. We searched the MEDLINE, EMBASE, and Cochrane databases for relevant research articles published from 2005 to 2021. The pooled hazard ratio (HR) with confidence interval (CI) was calculated to investigate the relationship between PARP expression and survival in multiple solid cancers. In total, 10,667 patients from 31 studies were included. A significant association was found between higher PARP expression and overall survival (OS) (HR = 1.54, 95% CI = 1.34–1.76, *p* < 0.001), disease-free survival (DFS) (HR = 1.15, 95% CI = 1.10–1.21, *p* < 0.001), and progression-free survival (PFS) (HR = 1.05, 95% CI = 1.03–1.08, *p* < 0.001). Subgroup analyses showed that PARP overexpression was significantly related to poor OS in patients with breast cancers (HR = 1.38, 95% CI = 1.28–1.49, *p* < 0.001), ovary cancers (HR = 1.21, 95% CI = 1.10–1.33, *p* = 0.001), lung cancers (HR = 2.11, 95% CI = 1.29–3.45, *p* = 0.003), and liver cancers (HR = 3.29, 95% CI = 1.94–5.58, *p* < 0.001). Regarding ethnicity, Asian people have almost twice their worst survival rate compared to Caucasians. The pooled odds ratio analysis showed a significant relationship between higher PARP expression and larger tumour size, poor tumour differentiation, lymph node metastasis, distant metastasis, higher TNM stage and lymphovascular invasion, and positive immunoreactivity for Ki-67, BRCA1, and BRCA2. In addition, nuclear expression assessed by the QS system using Abcam and Santa Cruz Biotechnology seems to be the most commonly used and reproducible IHC method for assessing PARP expression. This meta-analysis revealed that higher PARP expression was associated with a worse OS, DFS, and PFS in patients with solid cancers. Moreover, inhibition of this pathway through its specific inhibitors may extend the survival of patients with higher PARP expression.

## 1. Introduction

Poly (ADP-ribose) polymerases (PARPs) are DNA damage repair proteins, and their inhibitors have received great attention from researchers owing to their promising results in clinical trials [1,2]. PARPs are generally involved in many biological activities, including cell proliferation and cell death, mRNA transcription, DNA replication and repair, inflammation, cell apoptosis, and the maintenance of genomic integrity [1,2]. Upon DNA damage, PARPs bind to the damaged site and produce a poly-ADP chain from the NAD+ substrate that recruits the DNA repair protein through single-stranded break (SSB) or double-stranded break (DSB) repair pathways [1,2]. The overexpression of PARPs has been examined in various cancers and is linked with a resistance to DNA-damaging therapeutic agents [2,3]. The blockade of this pathway by specific PARP inhibitors inhibits the recruitment of DNA repair proteins and causes cell death, which may extend the long-term survival of patients with cancer [2,4]. The efficacy of PARP inhibitors is currently being investigated in clinical trials. Olaparib is the first US Food and Drug Administration (FDA)-approved PARP inhibitor for use in treating advanced ovarian cancer with germline BRCA1/2 mutations [5,6]. In a phase 2 clinical trial, Olaparib treatment at a dose of 400 mg twice/day in platinum-sensitive and high-grade ovarian cancer patients significantly improved their median progression-free survival (PFS) by 4 months [6].

Based on this background, many studies have investigated the prognostic significance of PARP expression in various cancers, and their results were inconsistent because of their limited sample sizes and suboptimal study designs [7,8,9,10,11,12,13,14,15,16,17,18,19,20,21,22,23,24,25,26,27,28,29,30,31,32,33,34,35,36,37]. The prognostic significance of PARPs has been studied in multiple cancers, such as breast [10], ovarian [24], lung [26], glial [32], oesophageal [33], and pancreatic cancers [35]. Most studies in these cancers have shown that a higher expression of PARP-1 is associated with poor outcomes [8,9,10,11,12,13,14,15,16,17,18,19,20,21,22,23,24,25,26,27,28,29,30,31,32,33,34,36,37]. However, a better prognosis has been reported in a few studies on pancreatic [35] and breast cancers [7]. The discrepancy between these studies may be due to the study design, sample size, organ type, ethnicity, and other secondary factors. Hence, it is indispensable to evaluate the prognostic significance of PARPs in various solid cancers.

In the present study, we conducted a systematic review and meta-analysis to reveal the prognostic value of PARP expression in solid cancers and to further correlate with the clinicopathological and immunohistochemical (IHC) markers. Additionally, we analysed the implications of the antibody clones, scoring systems, and the localisation of the immunoreactivity in IHC for the PARPs.

## 2. Materials and Methods

We performed a systematic review and meta-analysis of the literature according to the following guidelines set out by the PRISMA (Preferred Reporting Items for Systematic Reviews and Meta-analysis) statement [38]. The detailed protocol for this systematic review is registered in PROSPERO (280990).

### 2.1. Search Strategy

The Institutional Review Board of the Catholic University of Korea, College of Medicine approved this meta-analysis-based study (SC20ZISE0050). For this study, we retrieved articles between January 2005 and January 2021 from major electronic databases such as PubMed (MEDLINE), EMBASE, and the Cochrane Library using the following search keywords: “poly(ADP-ribose) polymerases”, “neoplasms”, “PARP-1 prognostic significance”, “immunohistochemistry”, and “PARP malignant neoplasm”. We also manually obtained articles to identify relevant papers. These records were managed using EndNote 20 (ver. 20.0.1, Bld. 15043, Thomson Reuters, New York, NY, USA).

### 2.2. Inclusion and Exclusion Criteria

Studies were included if they met the following criteria: (1) studies that included solid cancers, (2) studies that measured the PARP expression in tumour tissues via IHC and polymerase chain reaction (PCR), and (3) studies that showed the association between PARP expression and the clinicopathological parameters and survival outcomes, such as overall survival (OS) and disease-free survival (DFS). In the exclusion process, the following criteria were used: (1) meta-analyses, reviews, duplicate reports, commentaries, ongoing studies, editorials, non-English papers, or conference abstracts and (2) articles whose experimental designs and research methods were not similar and were mainly conducted on cell lines and animals.

### 2.3. Data Extraction and Quality Assessment

The author NT independently extracted the relevant data from the included studies and noted any discrepancies, which were resolved by consulting YC. The following data were extracted from each eligible study: name of the first author, year of publication, country, study period, organ, patient age range, median age, median follow-up period, PARP phenotype, chemotherapy regimen used, detection method used, PARP cut-off value, results of the survival analysis, and HRs. The information about IHC such as antibody vendors and clones, scoring systems, and localisation of the immunoreactivity (nuclear/cytoplasmic) were also documented. Furthermore, we used the Newcastle–Ottawa quality assessment scale (NOS) to evaluate the quality of all the eligible articles. The NOS consists of three parameters: selection, comparability, and outcome, with the total scores ranging from 0 to 9.

### 2.4. Statistical Methods

For the statistical analysis, we used Review Manager (version 5.3; Cochrane Collaboration, Oxford, UK) to calculate the pooled hazard ratios (HRs) with a 95% confidence interval (95% CI) by making a forest plot and investigated the relationship between PARP expression and patient survival (direct method). A subgroup analysis was conducted to determine the heterogeneity using a fixed-effects model, with HRs >1 denoting adverse outcomes. In the studies with Kaplan–Meier (K–M) curves but no HRs, the method used by Parmar et al. was performed to extract the data from the K–M curves and calculate the HR (indirect method) [39]. Mantel–Haenszel pooled odds ratios (ORs) with 95% CIs were used to evaluate the relationship between the PARP expression and clinicopathological and other IHC markers. An I^2^ statistic greater than 50% indicated the existence of heterogeneity between the studies. Begg and Egger’s tests were used to evaluate the publication bias quantitatively. Statistical significance was set at *p* < 0.05.

## 3. Results

### 3.1. Search Results

Figure 1 summarises the flowchart of the article selection process for the meta-analysis and review. The initial database searches identified a total of 5589 records (2373 in MEDLINE, 3113 in EMBASE, and 90 in the Cochrane library), and 13 supplementary records were identified from forward and backward searches, all of which were then imported to Endnote. After removing 780 duplicate records, 2939 records were removed because of irrelevant reference types. Next, 1247 records were excluded by titles, and the abstracts from 653 records were reviewed. After the exclusion of 510 records by abstract reviewing, a full-text review was performed for 143 records. After excluding 112 records, only 31 records (consisting of 33 cohorts) met the inclusion criteria for qualitative and meta-analytic synthesis.

### 3.2. Study Characteristics

The major characteristics of all the studies included in this meta-analysis are summarised in Table 1. In total, 31 records comprising 33 cohorts and 10,667 patients were included. Most of the studies were conducted in China (*n* = 8), followed by the UK (*n* = 4); Hungary and South Korea (*n* = 3 each); Egypt, Italy, Germany, the USA, and France (*n* = 2 each); and Poland, Spain, Saudi Arabia, Japan, and Austria (*n* = 1 each) (Table 1) [7,8,9,10,11,12,13,14,15,16,17,18,19,20,21,22,23,24,25,26,27,28,29,30,31,32,33,34,35,36,37]. The ages of the patients ranged from 21 to 89 years. The sample sizes of all the studies ranged from 50 to 2811, and the follow-up periods ranged from 1.9 to 15 years. The studies included in this meta-analysis were published between 2010 and 2021. The PARP phenotypes included in this study were mostly PARP-1 (*n* = 23) with the occasional PARP-2 (*n* = 1) and PARP-3 (*n* = 1), and the detection methods used to evaluate the PARP expression levels were IHC (*n* = 28) and PCR (*n* = 3). Moreover, 11 cohorts reported OS, five cohorts reported DFS, two cohorts reported PFS, nine cohorts reported both DFS and OS, and six cohorts reported both PFS and OS. The following cohorts were stratified according to organ types: breast (*n* = 14) [7,8,9,10,11,12,13,14,15,16,18,19], ovary (*n* = 7) [20,21,22,23,24,25], lung (*n* = 3) [26,27,28], liver (*n* = 2) [29,30], soft tissue (*n* = 2) [31,37], brain (*n* = 1) [32], oesophagus (*n* = 1) [33], stomach (*n* = 1) [34], pancreas (*n* = 1) [35], and skin (*n* = 1) [36]. The NOS scores of all the studies were higher than 7, which represents a relatively good quality (Table 1 and Appendix A).

Detailed information about the IHC scoring methods and the cut-off values used for PARP expression in the included studies are summarised in Table 2 and Appendix A. Tissue microarrays and whole tissue sections were commonly used (*n* = 14 each). Most studies used PARP antibodies from Abcam (*n* = 8; Cambridge, UK) and Santa Cruz Biotechnology (*n* = 7; Dallas, TX, USA). To interpret the IHC results, the H score (*n* = 10), immunoreactivity score (IRS, *n* = 6), and quick scoring system (QS, *n* = 5) were highly exploited. Nuclear location (*n* = 26) was mainly used for immunoreactivity (Table 2).

### 3.3. Association between PARP Expression and OS

Twenty-four studies, including twenty-six cohorts, reported an association between PARP expression and OS in 8471 patients with cancer. In a pooled HR analysis using the fixed model, a higher expression of PARPs was significantly associated with reduced OS in all cancers (HR = 1.54, 95% CI: 1.34–1.76, *p* < 0.001) (Figure 2). In a subgroup analysis, a higher expression of PARPs was significantly associated with poor OS in breast cancers (HR = 1.38, 95% CI = 1.28–1.49, *p* < 0.001), ovary cancers (HR = 1.21, 95% CI = 1.10–1.33, *p* < 0.001), lung cancers (HR = 2.11, 95% CI = 1.29–3.45, *p* = 0.003), and liver cancers (HR = 3.29, 95% CI = 1.94–5.58, *p* < 0.001) (Figure 3A). A quantitative synthesis of two soft tissue studies was not done due to the differences of the histologic subtypes of both studies. No significant relationship was found in miscellaneous cancers (Figure 3A). 

Regarding ethnicity, poor survival was associated with a higher expression of PARPs across different groups with almost double the HR in Asians than Caucasians (Asian: HR = 2.37, 95% CI = 1.86–3.02, *p* < 0.001; Caucasian: HR = 1.24, 95% CI = 1.18–1.30, *p* < 0.001) (Figure 3B). 

OS based on both univariate and multivariate analyses showed a significant association with adverse survival (univariate: HR = 1.10, 95% CI = 1.06–1.14, *p* < 0.001; multivariate: HR = 1.60, 95% CI = 1.46–1.76, *p* < 0.001) (Appendix A). Regarding the sample size, only studies with more than 100 samples showed a significant association with poor OS (HR = 1.26, 95% CI = 1.20–1.33, *p* < 0.001) (Appendix A). Both direct and indirect methods (pooled HRs versus K–M curve data extraction) showed a correlation with poor OS (direct: HR = 1.58, 95% CI = 1.33–1.88, *p* < 0.001; indirect: HR = 1.41, 95% CI = 1.12–1.78, *p* < 0.001) (Appendix A).

Based on a chemotherapy regimen, a significantly poor OS was found with high PARP expression in breast cancer patients receiving neoadjuvant chemotherapy of anthracycline and taxane (HR = 1.98, 95% CI = 1.04–3.78 *p* = 0.04) and ovarian cancer patients receiving an adjuvant chemotherapy of paclitaxel and carboplatin (HR = 1.15, 95% CI = 1.03–1.28, *p* = 0.01) and platinum-based chemotherapy (agents not specified) (HR = 1.52, 95% CI = 1.15–2.02, *p* = 0.003) (Appendix A).

According to antibody types, only Abcam (HR = 1.68, 95% CI = 1.19–2.36, *p* = 0.003) and Santa Cruz Biotechnology (HR = 1.92, 95% CI = 1.44–2.56, *p* < 0.001) showed significantly poor OS with high PARP expression (Figure 4A). Both nuclear and combined nuclear and cytoplasm immunoreactivity showed a significant association with poor OS (nuclear: HR = 1.57, 95% CI = 1.32–1.87, *p* < 0.001; combined nuclear and cytoplasm: HR = 3.54, 95% CI = 2.04–6.16, *p* < 0.001) (Figure 4B).

Finally, according to the scoring methods, a significant poor OS was associated with the H score system (HR = 1.56, 95% CI = 1.20–2.20, *p* < 0.001), QS (HR = 1.58, 95% CI = 1.40–1.78, *p* < 0.001), staining intensity (SI) (HR = 2.61, 95% CI = 1.12–6.09, *p* = 0.03), Allred score (HR = 3.65, 95% CI = 1.88–7.06, *p* = 0.0001), and computer-based scoring (HR = 1.82, 95% CI = 1.32–2.51, *p* < 0.001) (Figure 5).

### 3.4. Association of High PARP Expression with DFS and PFS

Fourteen studies reported DFS, and seven studies, including eight cohorts reporting PFS, were included in the meta-analysis (Figure 6). High PARP expression was associated with poor DFS (HR = 1.15, 95% CI = 1.10–1.23, *p* < 0.001). Similarly, high PARP expression was associated with significantly worse PFS (HR = 1.05, 95% CI = 1.03–1.08, *p* < 0.001) (Figure 6). PARP expression was associated with poor DFS in breast cancers and poor PFS in ovary cancers (Appendix A). A subgroup analysis according to chemotherapy regimen, univariate analysis and multivariate analysis, ethnicity (Asian vs. Caucasian), and direct/indirect methods (pooled HRs vs. K–M curve data extraction) showed a significant relationship with a poor outcome (Appendix A).

### 3.5. Association of High PARP Expression with Clinicopathological Parameters and Immunohistochemical Markers

The association of PARPs with the clinicopathological parameters and IHC markers is summarised in Table 3. Higher PARP expression was significantly associated with larger tumours (OR = 1.53, 95% CI = 1.00–2.34, *p =* 0.048), a higher tumour grade (HR = 1.53, 95% CI = 1.00–2.34, *p =* 0.048), the presence of lymph node metastasis, the presence of distant metastases (OR = 1.53, 95% CI = 1.00–2.34, *p =* 0.048), a higher TNM stage (HR = 1.53, 95% CI = 1.00–2.34, *p =* 0.048), and the presence of lymphovascular invasion (Table 3 and Appendix A). However, no significant relationship was found between age and T stage (Table 3 and Appendix A).

The pooled results showed that high PARP expression was significantly associated with the positivity of Ki-67 (OR = 0.12, 95% CI = 0.08–0.19, *p =* 0.048), BRCA1 (OR 0.12, 95% CI 0.08, 0.19, *p =* 0.048), and BRCA2 (OR 2.78, 95% CI (1.94, 3.98, *p =* 0.048) (Table 3 and Appendix A).

### 3.6. Publication Bias

Table 4 summarises the publication bias assessments. The funnel plot showed asymmetry in the OS, DFS, and PFS, and the trim-and-fill method was used to create a symmetrical funnel (Appendix A). Publication bias was significantly associated with the OS, DFS, and age, according to the Egger test (*p* = 0.024, *p* = 0.001, and *p* = 0.017, respectively). In contrast, none of the parameters showed any publication bias according to the Begg test (Table 4).

## 4. Discussion

The present comprehensive systematic review and meta-analysis is the first to scrutinise the prognostic value and clinicopathological significance of PARP expression in patients with solid cancers. First, high PARP expression was associated with poor OS, DFS, and PFS in most cancers, as previously reported. Second, by ethnicity, the Asian population showed a higher HR than the Caucasian subgroup (HR = 2.37 vs. 1.24), which means a high susceptibility of the Asian population to high PARP expression. Third, high PARP expression was associated with clinicopathological prognostic markers and positive immunoreactivity for Ki-67, BRCA1, and BRCA2. Fourth, according to the IHC clones, localisation methods, and scoring systems, nuclear expression by Abcam and Santa Cruz Biotechnology antibody clones were most commonly used, and the QS method was most reproducible, with the lowest heterogeneity.

In this study, we confirmed a significantly poor OS of high PARP expression in breast, ovary, lung, and liver cancers. In a previous meta-analysis on breast cancers by Qiao et al., the authors reported a significantly poor OS of high PARP expression in early breast cancer but not in locally advanced breast cancers [40]. In contrast, through our systematic analysis, we found that many studies other than Aiad et al.’s included stage III or IV breast cancers (Table 1) [7]. However, Qiao et al. did not include these studies in the locally advanced breast cancer group in their subgroup analysis but included them in the nonlocally advanced breast cancer group [40]. The results of Aiad et al.’s study should be interpreted with caution, because it included only stage IIIB breast cancers, only 84 cases, which is relatively small, and only core needle biopsy samples (no other study used core needle biopsy samples), which could highly result in a sampling bias [7]. In addition, Qiao et al. somehow combined two different results from one cohort of Aiad et al.’s study to perform a subgroup analysis of a locally advanced breast cancer group [40]. The one result was targeting cytoplasmic immunoreactivity while another was targeting nuclear immunoreactivity, which finally misled to the wrong conclusion that high PARP expression is associated with better survival in locally advanced breast cancers and poor survival in early breast cancers. In this study, other than that, we included all results from the additional studies and found a higher statistical significance of high PARP expression to poor OS in overall breast cancers. Similarly, the same relationship between PARP expression and poor OS was found in ovary, lung, and liver cancers.

On the other hand, additional studies with more samples are still required in other cancers, such as pancreas, brain, soft tissues, skin, and stomach, to investigate the exact prognostic role of PARPs. Interestingly, a better survival of high PARP expression was reported in pancreatic cancers [35], even though it was only one study, which raises the possibility of the different roles of PARPs in different cancers. Therefore, more studies with sufficient samples are expected in these cancers.

According to the quality assessment, heterogeneity was high in the breast, ovary, and miscellaneous cancer groups (I^2^ = 71%, 61%, and 77%, respectively), although the NOS of the included studies were relatively good. The source of heterogeneity may be due to different cancer types, histological subtypes, stages, IHC scoring systems, cut-off values, antibody clones, target PARP phenotypes, etc. Additionally, there seemed to be a publication bias among the included studies. However, the subgroup analysis also consistently revealed a statistically significant relationship of PARP expression to a poor prognosis, regardless of the cancer types, ethnicity, statistical analysis methods (univariate vs. multivariate), direct/indirect methods for HR extraction, antibody clones, scoring systems, and localisation of immunoreactivity. In addition, similar findings were found in DFS and PFS in every subgroup showing an unfavourable impact of PARP expression in solid cancers. These are consistent findings reported in many hematologic disorders (nonsolid cancers) as well [41].

The subgroup analysis by ethnicity revealed that Asian patients have a worse survival rate compared to Caucasians. Our results were also consistent with the previous findings of two meta-analyses, which reported that the allele frequency of A in the PARP-1 V762A polymorphism is significantly higher in Asian patients compared to Caucasians and related to a higher risk of cancer development [42,43]. The difference between both ethnicities is due to their genetic backgrounds and environmental variables. These result in tumours with various biological characteristics [44]. Furthermore, the most common tumour sites within organs differ between Asian and Caucasian populations, resulting in variations in tumour activity and prognoses [45]. This could be the reason for the differences in HR between Asians and Caucasians. Overall, our results indicate that Asian patients have a higher risk of cancer with a higher PARP expression than Caucasians.

Next, a significant relationship was found between a higher PARP expression and various clinicopathological risk factors, such as tumour size, tumour differentiation, lymph node metastasis, distant metastasis, TNM stage, and lymphovascular invasion. These findings indicate that PARPs are involved in cancer development, proliferation, invasion, and metastasis, which are mainly related to a poor prognosis and cancer mortality in patients. At the cellular level, the main effect of PARPs is to regulate the proliferation, migration, and invasion of cancer cells [46,47,48]. A previous study reported that PARPs activate the metastasis-related genes integrin β-1, matrix metallopeptidase-2 (MMP-2), and MMP-9 through the nuclear factor kappa-light-chain pathway in colon cancer [46]. Another study showed that a loss of function in PARP-1 activated the epithelial–mesenchymal transition pathway through the increased expression of N-cadherin and ZEB-1 and the decreased expression of E-cadherin and β-catenin, which promote tumour progression through the TGF-β signalling pathway [47].

Furthermore, Ki-67, BRCA1, and BRCA2 expressions were also significantly associated with a higher PARP expression. In primary mucosal melanoma, it has been observed that a higher PARP-1 expression was significantly associated with higher mitotic activity, which showed that PARP-1 is involved in the regulation of mitosis. Moreover, a loss of function in BRCA1 and BRCA2 most commonly occurs in breast and ovarian cancers [49]. It has been observed that breast tumours with BRCA1/2 mutations lack homologous repair recombination capabilities, making it difficult to repair DNA damage, causing cell apoptosis. [50]. A previous study demonstrated that a higher expression of PARP-1, BRCA 1, and BRCA2 resulted in a shorter OS at 10 years in patients with breast cancer [14]. Hence, there may be the possibility that these patients might benefit from treatment with PARP inhibitors. Emerging evidence has revealed that BRCA1 mutations in breast and ovarian cancers are more sensitive to PARP inhibitors [51].

Interestingly, we found that only Abcam and Santa Cruz Biotechnology showed poor OS with high PARP expression in subgroup analyses by antibody clones. The heterogeneity in Abcam was higher than Santa Cruz Biotechnology (I^2^ = 71% vs. 41%) (Figure 4A). In terms of immunoreactivity location-wise, nuclear localisation was commonly targeted as compared to the cytoplasm and combined nuclear and cytoplasm. Although the heterogeneity in nuclear localisation was significantly higher than the combined nuclear and cytoplasm (I^2^ = 70% vs. 0%), both represent a poor OS with high PARP expression (Figure 4B). The primary reason for a higher heterogeneity may be different antibody clones, scoring systems, and cancer types. According to the scoring system, the QS method was more reproducible as compared to other scoring methods, such as the H score, IRS, and SI-based scoring methods (I^2^ = 0%, 31%, 76%, and 86%, respectively) (Figure 5). Based on this data, we concluded that the QS method is easier and uniformly applicable compared to other scoring methods. Further studies should be designed based on these findings.

PARP inhibitors have emerged as a promising target-based therapy owing to their excellent results in clinical trials [52,53,54,55]. Among 17 members of the PARP family, PARP-1 is the most well-studied and has been identified as a key player in the DNA repair pathway [52,53,54,55]. Generally, PARPs have three functional domains: a DNA-binding domain that comprises zinc finger motifs, an auto-modification domain, and a carboxyl catalytic domain [52,53,54,55]. Damaged DNA is often repaired via two pathways, SSB and DSB, which are critical for maintaining cell viability [52,53,54,55]. SSBs are repaired via the mismatch repair, base excision repair (BER), and nucleotide-excision repair pathways, while DSBs are repaired through the homologous recombination and nonhomologous end-joining repair pathways [52,53,54,55]. If any damage occurs on the DNA strand, PARPs bind to the DNA damage site and produce a poly-ADP chain from the NAD+ substrate that recruits the DNA repair protein through the SSB or DSB repair pathway [52,53,54,55].

In our study, a subgroup analysis according to the chemotherapy regimen in breast and ovarian cancer patients showed a poor OS and PFS in the high PARP expression group regardless of the chemotherapy regimen. Although this alone cannot be direct evidence of the adverse biologic role of PARP expression because we could not include chemotherapy-free cohorts, the results blend smoothly with the other findings from many other preclinical and clinical trial study results [52,53,54,55]. A previous preclinical study showed that treatment with PARP inhibitors leads to cell cycle arrest and cell death in cells with a homologous recombination deficiency (HRD) [54,56,57]. In the pancreas, a clinical trial treating metastatic pancreatic cancer patients with germline BRCA1/2 mutations with Olaparib extended the PFS [58]. In the ovaries, a clinical trial showed that using Olaparib, paclitaxel, and carboplatin increased the PFS in recurrent cancer patients [59]. All these lines of evidence indirectly support our results. To overcome the aforementioned limitations, more evidence on the effects of the treatment regimen is needed in the future.

Robust efforts have been made to perform this comprehensive meta-analysis. However, our study has some limitations, such as: (i) studies not published in the English language were excluded due to the hassle of obtaining the data more precisely, which may have caused a bias in our results; (ii) in the case of studies without HRs with 95% CI, the data were extracted using digitised software from the K–M curve (indirect method) before the pooled HR was calculated, which may have compromised the accuracy of the data; (iii) the cut-off values differentiating between high and low PARP expression observed via IHC varied among the included studies, which may be the source of heterogeneity in the overall results and subgroup results; and (iv) the different clones and sources of antibodies used for PARP expression between the studies may have also impacted the precise estimation of the prognoses. Therefore, a large multicentre study using the same cut-off values with the same clones and sources of antibodies is required to obtain more precise outcomes. Regardless of the above limitations, our meta-analysis revealed the prognostic and clinicopathological significance of PARPs in solid cancers.

## 5. Conclusions

The present meta-analysis revealed that a higher expression of PARPs could act as a risk factor for poor OS, DFS, and PFS in various solid cancers, such as breast, ovary, lung, and liver cancers. Asians with a high PARP expression are more susceptible to a worse survival diagnosis than Caucasians. Furthermore, a high PARP expression was significantly associated with aggressive clinicopathological parameters and positive immunoreactivity for Ki-67, BRCA1, and BRCA2. In addition, nuclear expression assessed by the QS system using Abcam and Santa Cruz Biotechnology seems to be the most commonly used and reproducible IHC method for assessing PARP expressions. Collectively, the inhibition of this pathway through its specific inhibitors may extend the survival of patients with high PARP expression. Well-designed multicentre studies with larger sample sizes are required in miscellaneous cancers, such as the pancreas.

## Figures and Tables

**Figure 1 cancers-13-05594-f001:**
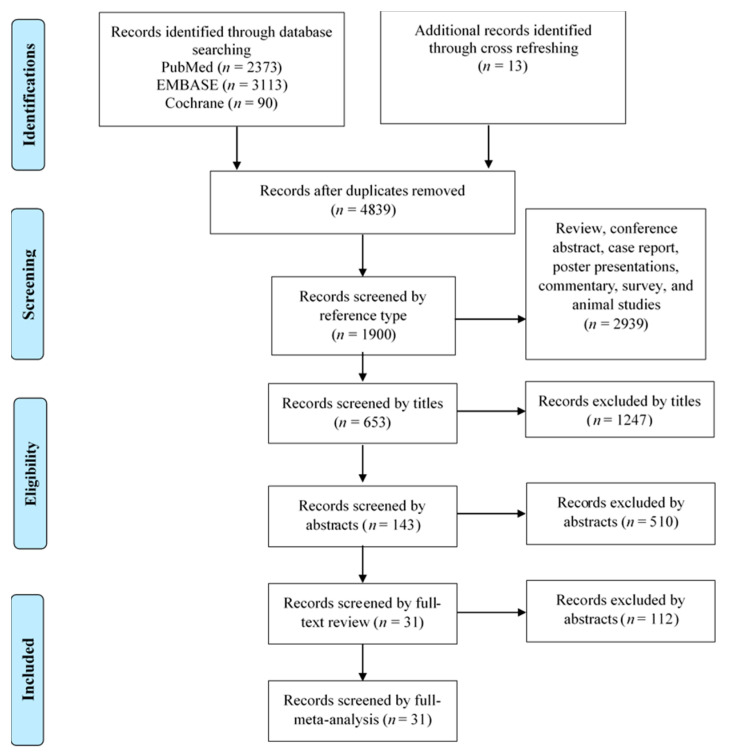
Flow chart for the study selection process.

**Figure 2 cancers-13-05594-f002:**
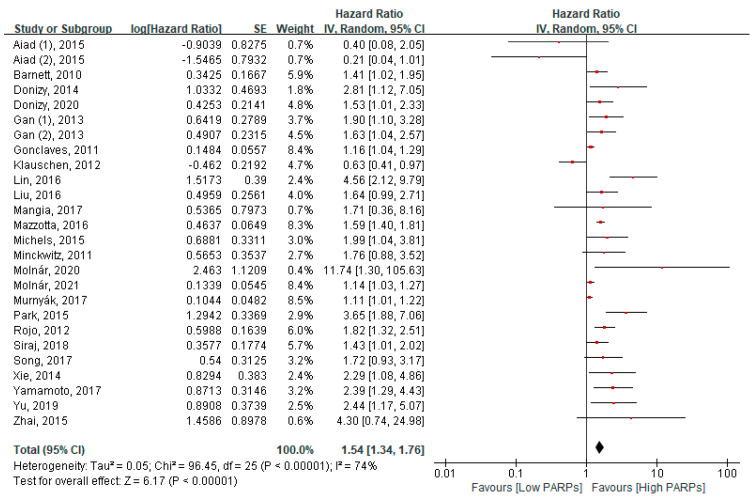
Forest plot of the hazard ratio for the PARP expression and overall survival in solid cancers.

**Figure 3 cancers-13-05594-f003:**
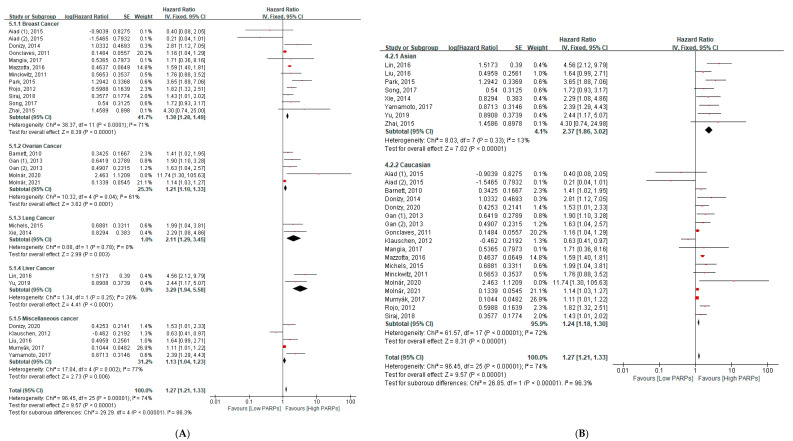
The subgroup analysis between the PARP expression and overall survival (**A**) according to the types of cancers and (**B**) ethnicity.

**Figure 4 cancers-13-05594-f004:**
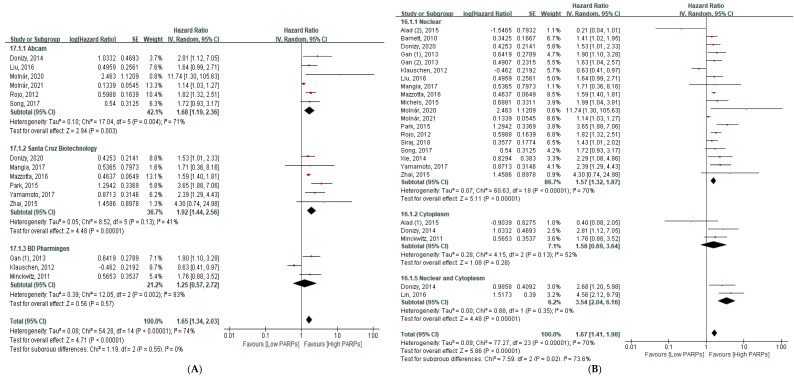
The subgroup analysis between the PARP expression and overall survival (**A**) according to the antibody clones and (**B**) immunoreactivity location-wise.

**Figure 5 cancers-13-05594-f005:**
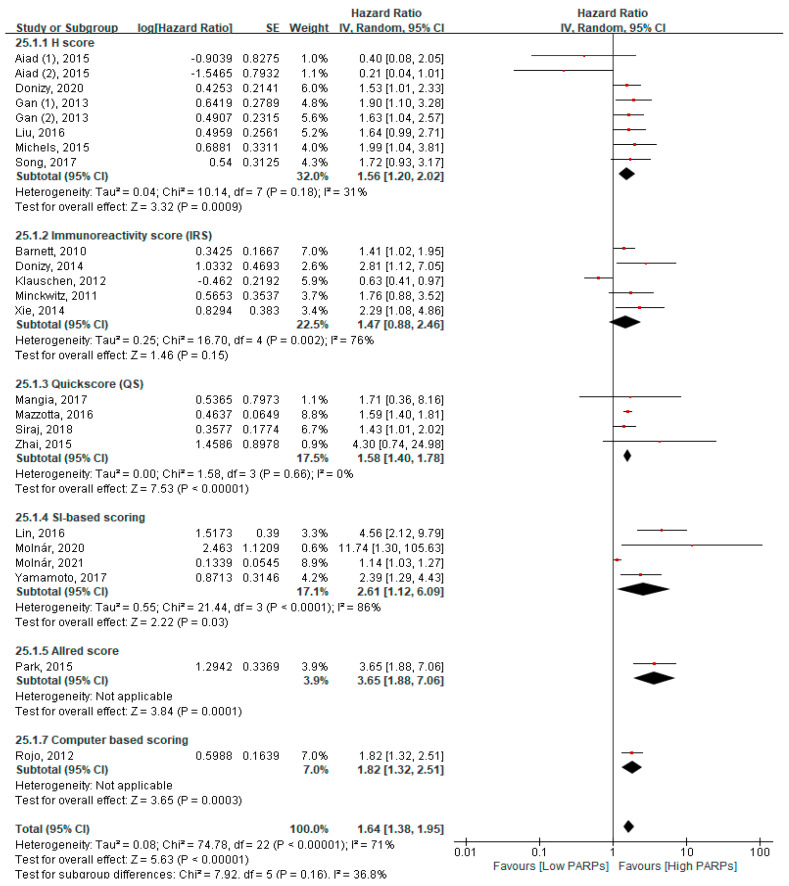
The subgroup analysis between the PARP expression and overall survival according to the scoring methods.

**Figure 6 cancers-13-05594-f006:**
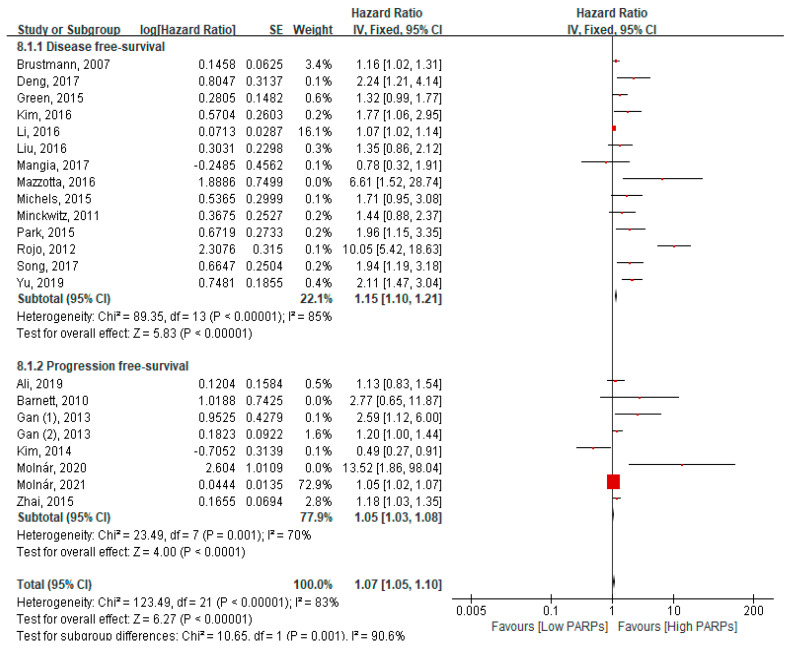
Forest plot of the hazard ratio for PARP expression and disease-free survival and progression-free survival in solid cancer patients.

**Table 1 cancers-13-05594-t001:** Basic characteristics of all the included studies regarding PARPs for this meta-analysis.

Organ	Authors	Year	Country	Ethnicity	Study Period	Cancer Type/TNM Stage	Patients No.	Average Age (Range)	Median f/u (Years)	PARP Phenotype	Chemo Regimen	Detection Method	Hazard Ratio (CI 95%)	NOS Score
Breast	Gonclaves [10]	2011	France	Caucasian	NA	IDC/ILC/MC/ others Stage I-IIIB	2485	NA	8	PARP-1	Adj. Ant/Tax/CMF	PCR	OS: 1.16 (1.04–1.29)	8
	Minckwitz [16]	2011	Germany	Caucasian	2001–2005	IDC/ILC/ others Stage IIA-IIIB	638	NA	4.8	PARP	Neoadj. Ant/Tax	IHC	OS: 1.76 (0.87–3.50) DFS: 1.44 (1.88–2.36)	8
	Rojo [17]	2012	Spain	Caucasian	1998–2000	IDC/ILC/ others NA	330	58 (26–90)	8.5	PARP-1	Adj. CMF/horm	IHC	OS: 1.82 (1.32–2.52) DFS: 10.05(5.42–18.66)	8
	Donizy [9]	2014	Poland	Caucasian	1993–1994	IDC Stage IIA-IIB	83	55.2	15	PARP-1	Neoadj. CMF/Ant	IHC	OS: 2.81 (1.12–7.03)	7
	Aiad (1) * [7]	2015	Egypt	Caucasian	2008–2012	IDC Stage IIIB	84	53 (29–86)	NA	PARP-1	Neoadj.FEC	IHC	OS: 0.40 (0.08–2.05)	7
	Aiad (2) * [7]	2015	Egypt	Caucasian	2008–2012	IDC Stage IIIB	84	53 (29–86)	NA	PARP-1	Neoadj.FEC	IHC	OS: 0.21 (0.04–1.00)	7
	Green [11]	2015	UK	Caucasian	1989–2004	OBC Stage I–III	1269	55	NA	PARP-1	NA	IHC	DFS: 1.32 (0.99–1.77)	7
	Park [14]	2015	South Korea	Asian	1997–2003	IC/ILC Stage I–IV	192	47 (22–73)	11.2	PARP-1	Adj. Ant/Tax/CMF	IHC	OS: 3.64 (1.88–7.05) DFS: 1.95 (1.44–3.34)	8
	Zhai [15]	2015	China	Asian	2007–2012	IDC/ILC Stage I-IIB	198	53 (29–70)	4	PARP-1	Neoadj. Ant/Tax	IHC	OS: 4.30 (0.74–25.00) PFS: 1.18 (1.03–1.35)	8
	Mazzotta [13]	2016	Italy	Caucasian	1998–2012	IDC/others Stage I-IIIB	114	53	4.8	PARP-1	Adj. Horm/chemo	IHC	OS: 1.59 (1.40–181.19) DFS: 6.61 (1.52–28.80)	8
	Deng [8]	2017	China	Asian	2000–2012	TNBC Stage I-IIIB	118	51.6 (25–81)	6.2	PARP-1	Adj. Ant/Tax	IHC	DFS: 2.23 (1.20–4.13)	8
	Mangia [12]	2017	Italy	Caucasian	1996–2012	IDC/ILC/ others NA	308	51 (24–80)	6.1	PARP-1	NA	IHC	OS: 1.71 (0.36–8.16) DFS: 0.78 (0.32–1.91)	8
	Song [19]	2017	China	Asian	2005–2010	IDC, LC, MC Stage I-IV	547	51 (20–82)	9.8	PARP-3	Adj. CAF/CEFD	IHC	OS: 1.71 (0.93–3.15) DFS: 1.94 (1.19–3.19)	8
	Siraj [18]	2018	Saudi Arabia	Caucasian	1990–2011	IDC, ILC, MC Stage I–IV	1008	45 (39–54)	4	PARP	NA	IHC	OS: 1.43 (1.01–2.04)	8
Ovary	Brustmann [22]	2007	Austria	Caucasian	1985–1996	SOC Stage I–III	50	64	NA	PARP	Adj. PBC	IHC	DFS: 1.16 (1.02–1.31)	7
	Barnett [21]	2010	USA	Caucasian	1995–2003	SOC Stage I–IV	186	61 (19–86)	NA	PARP	Adj. PBC	IHC	OS: 0.71 (0.50–0.99) PFS: 0.36 (0.09–1.51)	7
	Gan (1) * [23]	2013	UK	Caucasian	1991–2007	SOC Stage I–IV	174	61 (36–86)	NA	PARP-1	Adj. PBC	IHC	OS: 1.90 (1.10–3.20) PFS: 2.59 (1.12–6.00)	7
	Gan (2) * [23]	2013	UK	Caucasian	1991–2007	SOC Stage I–IV	174	61 (36–86)	NA	C-PARP-1	Adj. PBC	IHC	OS: 1.63 (1.04–2.57) PFS: 1.20 (1.00–1.44)	7
	Ali [20]	2019	UK	Caucasian	1997–2010	SOC Stage I–IV	525	NA	NA	PARP-1	Adj. PBC	IHC	PFS: 1.13 (0.83–1.54)	7
	Molnar [24]	2020	Hungary	Caucasian	2011–2017	SOC Stage IIIA-IIIB	86	57	2.7	PARP	Adj. Pac/Carbo	IHC	OS: 11.74 (1.30–105.63) PFS: 13.52 (1.86–98.04)	8
	Molnar [25]	2021	Hungary	Caucasian	2011–2019	SOC Stage IIIA-IIIB	104	57.9	2.8	PARP	Adj. Pac/Carbo	IHC	OS: 1.14 (1.03–1.27) PFS: 1.05 (1.02–1.07)	8
Lung	Kim [27]	2014	South Korea	Asian	2008–2012	SCLC Stage I–III	79	62	1.6	PARP-1	Neoadj. Eto/Cis/Carbo	IHC	PFS: 0.49 (0.26–0.91)	8
	Xie [28]	2014	China	Asian	2008–2010	NSCLC Stage I–IV	111	63 (43–81)	NA	PARP-1	NA	IHC	OS: 2.29 (1.08–4.85)	7
	Michels [26]	2015	France	Caucasian	1994–2002	NSCLC Stage I-II	225	64 (40–82)	10.03	PARP	NA	IHC	OS: 1.99 (1.04–3.76) DFS: 1.71 (0.95–3.62)	8
Liver	Lin [29]	2016	China	Asian	2005–2008	HCC Stage I–IV	145	45.4	NA	PARP-2	NA	IHC	OS: 4.56 (2.12–9.79)	7
	Yu [30]	2019	China	Asian	NA	HCC Stage I–IV	298	NA	NA	PARP-1	NA	PCR	OS: 2.43 (1.17–5.07) DFS: 2.11 (1.46–3.04)	7
Soft tissue	Li [31]	2016	China	Asian	1996–2012	SC NA	50	55.1 (24–90)	5.4	PARP-1	NA	IHC	DFS: 1.07 (1.02–1.14)	7
	Kim [37]	2016	South Korea	Asian	1998–2013	STS Stage I–IV	105	NA	15	PARP-1	Adj. chemo	IHC	DFS: 2.78 (1.70–4.55)	8
Brain	Murnyák [32]	2017	Hungary	Caucasian	2006–2014	Glioma Stage II-IV	135	60.5 (21–89)	NA	PARP-1	NA	PCR	OS: 1.11 (1.01–1.22)	7
Oesophagus	Yamamoto [33]	2017	Japan	Asian	1998–2011	SCC IA-IVB	86	NA	3.5	PARP-1	NA	IHC	OS: 2.39 (1.29–4.44)	8
Pancreas	Klauschen [35]	2012	Germany	Caucasian	NA	PDAC Stage I–IV	178	NA	NA	PARP	NA	IHC	OS: 0.63 (0.41–0.96)	7
Skin	Donizy [36]	2020	USA	Caucasian	1989–2018	MM Stage I–IV	192	65	1.9	PARP-1	NA	IHC	OS:1.53 (1.01–2.33)	8
Stomach	Liu [34]	2016	China	Asian	NA	GC stage I–IV	564	60 (29–82)	5.5	PARP-1	NA	IHC	OS: 1.64 (0.99–2.71) DFS: 1.35 (0.86–2.12)	8

Abbreviations: PARP: nuclear and cytoplasmic poly(ADP-ribose) polymerases; NOS: Newcastle–Ottawa scale; NA: not available; IHC: immunohistochemistry; PCR: polymerase chain reaction; HR: hazard ratio; OS: overall survival; DFS: disease free-survival; PFS: progression free-survival; IDC: invasive ductal carcinoma; IC: invasive carcinoma; ILC: invasive lobular carcinoma; MC: mucinous carcinoma; TNBC: triple-negative breast cancer; FEC: fluorouracil, epirubicin, and cyclophosphamide; Ant/Tax: anthracycline/taxane; Horm: hormonal therapy; CMF: cyclophosphamide, methotrexate, and 5-fluorouracil; Chemo: chemotherapy; CAF/CEFD: cyclophosphamide/doxorubicin or epirubicin/5-fluorouracil and docetaxel; C-PARP: cleaved PARP; SOC: serous ovarian carcinoma (include serous ovarian carcinoma, serous cystadenocarcinoma, endometrioid, clear cell carcinoma, mucinous cystadenocarcinoma, mixed, other, and unknown); Carbo/Cyclo: carboplatin and cyclophosphamide; Pac: Paclitaxel; PBC: platinum-based chemotherapy; SCLC: small cell lung cancer; Eto/Cis/Carbo: etoposide/cisplatin/carboplatin; NSCLC: non-small cell lung cancer; HCC: hepatocellular carcinoma; SCC: squamous cell carcinoma; PDAC: pancreatic ductal adenocarcinoma; MM: mucosal melanomas; SC: sacral chordoma; STS: soft tissue sarcoma; GC: gastric cancer; and Adj.: adjuvant. Note: ***** Aiad (1): target cytoplasmic PARP, Aiad (2): target nuclear PARP, Gan (1), Target Nuclear PARP-1, and Gan (2): target nuclear cleaved PARP-1.

**Table 2 cancers-13-05594-t002:** Immunohistochemistry detection methods used for the PARP expression in the included studies.

Organ	Study, Year	Format of Sampling	IHC Evaluation Method	Antibody					No. of Involved Pathologists	IHC Cut-Off Value	Localisation	Number of High PARP Cases (%)
				Company	Source	Type	Clone	Dilution				
Breast	Minckwitz, 2011 [16]	TMA	IRS	BD Pharmingen	Mouse	mAb	7D3-6	1:1500	2 *	6	Cytoplasm	151 (23.7)
	Rojo, 2012 [17]	TMA	Computer based scoring	Abcam	Mouse	mAb	A6.4.12	1:300	1	NA (29–133.094)	Nuclear	103 (31.2)
	Donizy, 2014 [9]	TMA	IRS	Abcam	Rabbit	pAb	ab6079	1:150	2	6	Cytoplasm and Combined N&C	48 (57.8) 35 (42.2).
	Aiad (1), 2015 [7]	NCBs	H score	eBioscience	Mouse	mAb	C2-10	1:100	3	70	Cytoplasm	40 (48.0)
	Aiad (2), 2015 [7]	NCBs	H score	eBioscience	Mouse	mAb	C2-10	1:100	3	10	Nuclear	64 (76.0)
	Green, 2015 [11]	TMA	H score	BD Pharmingen	Mouse	mAb	7D3-6	1:1000	2 *	10	Nuclear	524 (41.2)
	Park, 2015 [14]	TMA	Allred score	Santa Cruz Biotechnology	Mouse	mAb	F-2, sc-8007	1:100	2	13	Nuclear	78 (41.0)
	Zhai, 2015 [15]	WTS	QS	Santa Cruz Biotechnology	Mouse	mAb	F-2, sc-8007	1:300	7 *	10	Nuclear	59 (54.6)
	Mazzotta, 2016 [13]	WTS	QS	Santa Cruz Biotechnology	Mouse	mAb	F-2, sc-8007	1:500	2	10	Nuclear	68 (59.6)
	Deng, 2017 [8]	WTS	QS	Abcam	Rabbit	pAb	ab6079	NA	3 *	10	Nuclear	52 (44.1)
	Mangia, 2017 [12]	TMA	QS	Santa Cruz Biotechnology	Mouse	mAb	F-2, sc-8007	1:500	2	10	Nuclear	76 (28.9)
	Siraj, 2018 [18]	TMA	QS	Cell signaling	Rabbit	mAb	D64E10	NA	2 *	10	Nuclear	451 (44.7)
	Song, 2017 [19]	WTS	H score	Abcam	Rabbit	pAb	96601	1:100	2	57.5	Nuclear	234 (47.5)
Ovary	Brustmann, 2007 [22]	TMA	IRS	Novocastra	Mouse	mAb	NA	1:30	1	8	Nuclear	38 (76.0)
	Barnett, 2010 [21]	WTS	IRS	NeoMarkers Fremont	Mouse	mAb	A6.4.12	1:200	1	8	Nuclear	101 (54.0)
	Gan (1), 2013 [23]	TMA	H score	BD Pharmingen	Mouse	mAb	7D3-6	1:600	1	180	Nuclear	33 (22.0)
	Gan (2), 2013 [23]	TMA	H score	Abnova	Mouse	mAb	A6.4.12	1:600	1	75	Nuclear	117 (79.0)
	Ali, 2019 [20]	TMA	H score	Cell signaling	Rabbit	mAb	46D11	1:600	3 *	80	Nuclear	208 (51.9)
	Molnar, 2020 [24]	WTS	SI based scoring	Abcam	Rabbit	pAb	Ab6079330	1:500	1 *	1	Nuclear	45 (52.3)
	Molnar, 2021 [25]	WTS	SI based scoring	Abcam	Rabbit	pAb	Ab6079330	1:500	1 *	1	Nuclear	70 (67.3)
Lung	Kim, 2014 [27]	WTS	SP based scoring	Bethyl Laboratories Inc.	Rabbit	pAb	00279	1:200	1 *	3	Nuclear	33 (41.8)
	Xie, 2014 [28]	WTS	IRS	Biorbyt	Rabbit	pAb	NA	1:300	2	4	Nuclear	62 (55.9)
	Michels, 2015 [26]	WTS	H score	Merck	Mouse	mAb	10H	1:1500	3 *	145 *	Nuclear	49 (53.3)
Liver	Lin, 2016 [29]	WTS	SI based scoring	Raybiotech, Inc	Goat	NA	Q9UGN5	NA	NA	2	Nuclear	75 (51.7)
Soft tissue	Li, 2016 [31]	WTS	Summation based scoring	Abcam	Rabbit	pAb	ab6079	1:50	NA	3	Combined N&C	39 (78.0)
	Kim, 2016 [37]	TMA	Allred score	Santa Cruz Biotechnology	Mouse	mAb	F-2 sc-8007	1:100	3	10	Nuclear	64 (57.1)
Oesophagus	Yamamoto, 2017 [33]	WTS	SI based scoring	Santa Cruz Biotechnology	Mouse	mAb	F2, sc-8007	1:50	2	2	Nuclear	54 (62.8)
Pancreas	Klauschen, 2012 [35]	TMA	IRS	BD Pharmingen	Mouse	mAb	7D3-6	1:1000	2	3	Nuclear	138 (77.5)
Skin	Donizy, 2020 [36]	WTS	H score	Santa Cruz Biotechnology	Mouse	mAb	sc-74470 (B10)	1:50	1	200	Nuclear	72 (43.0)
Stomach	Liu, 2016 [34]	TMA	H score	Abcam	Rabbit	pAb	ab6079	1:200	2	175	Nuclear	266 (47.2)

Abbreviations: mAb: monoclonal antibody, pAb; polyclonal antibody TMA: tissue microarray, WTS: whole tissue section, CNBs: core needle biopsies, IRS: immunoreactivity score, QS: quick score, SI: staining intensity, SP: staining percentage, N&C: nuclear–cytoplasmic expression, and HPA: Human Protein Atlas database. *: Signifies the number of pathologists from the articles list.

**Table 3 cancers-13-05594-t003:** The association of PARP expression with the clinicopathological and immunohistochemical parameters.

Parameters	Number of Studies	Number of Patients	Pooled OR (95% CI)	*p*-Value	Heterogeneity		
					I^2^ (%)	*p*-Value	Model
Clinicopathological parameters							
Age (<50 vs. >50)	8	2977	0.96 (0.82–1.12)	0.59	44	0.08	Fixed
Tumour size (<5 cm vs. >5 cm)	8	2453	1.60 (1.32–1.95)	<0.001 *	80	< 0.001	Fixed
Histologic grade (1 + 2 vs. 3)	16	4927	1.79 (1.58–2.04)	<0.001 *	72	< 0.001	Fixed
T stage (1 + 2 vs. 3+4)	4	1460	1.26 (0.91–1.75)	0.16	14	0.32	Fixed
Lymph node metastasis (absent vs. present)	17	3743	1.27 (1.10–1.46)	0.001 *	42	0.03	Fixed
Distant metastasis (absent vs. present)	6	1668	2.40 (1.79–3.23)	<0.001 *	61	0.03	Fixed
TNM stage (I + II vs. III + IV)	6	2260	1.49 (1.24–1.79)	<0.001 *	0	0.94	Fixed
Lympho-vascular invasion (absent vs. present)	10	3141	1.22 (1.05–1.42)	0.01 *	62	0.004	Fixed
Immunohistological markers							
Ki-67 (negative vs. positive)	5	2529	1.60 (1.35–1.90)	<0.001 *	83%	< 0.001	Fixed
BRCA1 (negative vs. positive)	4	1546	1.63 (1.32–2.02)	<0.001 *	89%	< 0.001	Fixed
BRCA2 (negative vs. positive)	3	1040	2.78 (1.94–3.98)	<0.001 *	92%	< 0.001	Fixed

Abbreviations: CI: confidential interval, OR: odds ratio, and TNM: tumour node metastasis. *****: Significant *p*-value.

**Table 4 cancers-13-05594-t004:** The publication bias assessment of the included studies.

Parameters	No. of Studies/Cohorts	Begg Test (*p*-Value)	Egger Test (*p*-Value)	Model
PARPs and survival outcomes				
Overall Survival	24/26	0.133	0.024 *	Fixed
Disease-free survival	14	0.062	0.001 *	Fixed
Progression-free survival	7/8	0.536	0.455	Fixed
PARPs and clinicopathological parameters				
Age (>50 vs. <50)	8	0.063	0.017 *	Fixed
Tumour size (>5 cm vs. <5 cm)	8	1.000	0.723	Fixed
Histologic grade (1 + 2 vs. 3)	4	0.308	0.345	Fixed
T stage (1 + 2 vs. 3 + 4)	17	0.964	0.532	Fixed
Lymph node metastasis (absent vs. present)	6	1.000	0.804	Fixed
Distant metastasis (absent vs. present)	6	0.259	0.354	Fixed
TNM stage (I + II vs. III + IV)	10	1.000	0.513	Fixed
PARPs and Immunohistological markers				
Ki-67 (negative vs. positive)	5	0.220	0.113	Fixed
BRCA1 (negative vs. positive)	4	0.308	0.368	Fixed
BRCA2 (negative vs. positive)	3	1.000	0.370	Fixed

*: Significant *p*-value.

## Data Availability

The data presented in this study are available on request from the corresponding author (https://www.researchgate.net/profile/Yosep-Chong, accessed on 8 November 2021).

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
