# Peer review of "High Poly(ADP-Ribose) Polymerase Expression Does Relate to Poor Survival in Solid Cancers: A Systematic Review and Meta-Analysis"

_cancers, 2021, doi:10.3390/cancers13225594_

Round 1

Reviewer 1 Report

In the manuscript, Nishant Thakur and collegues analized the prognostic role of PARP proteins in different solid cancer subtypes using online available data of numerous studies. Although the idea of the manuscript as an high potential clinical relevance, the experimental design lacks of a fundamental point. The main weakeness of the study is the correlation of PARP expression with clinical parameters ( OS, DFS or PFS) of cancer patients not considering the clinicial interventions that have been applied. Indeed, the authors seem not to consider the biological role of PARP proteins in the response to different types of therapies. In particular, authors should consider that the therapeutic/prognostic value of PARP proteins varies significantly in the response to chemio-free therapies, standard chemotherapy regimens or those using high intensive chemotherapy. The authors should reanalyzed the data related to OS, DFS and PFS considering the different therapeutical regimens. 

Author Response

Reviewers 1 comments

In the manuscript, Nishant Thakur and colleagues analyzed the prognostic role of PARP proteins in different solid cancer subtypes using online available data of numerous studies. Although the idea of the manuscript as a high potential clinical relevance, the experimental design lacks of a fundamental point. The main weakness of the study is the correlation of PARP expression with clinical parameters (OS, DFS or PFS) of cancer patients not considering the clinical interventions that have been applied. Indeed, the authors seem not to consider the biological role of PARP proteins in the response to different types of therapies. In particular, authors should consider that the therapeutic/prognostic value of PARP proteins varies significantly in the response to chemo-free therapies, standard chemotherapy regimens or those using high intensive chemotherapy. The authors should reanalyze the data related to OS, DFS and PFS considering the different therapeutically regimens.

Answer: Thank you for your comment to modify our manuscript. We could analyze OS and PFS with PARP expression according to clinical interventions such as chemotherapy regimen as per your suggestion although the included studies are not many. We also added description about this result in the results and discussion parts.

  1. Overall survival
  2. Progression-free survival

Supplementary Figure 3. Forest plot of the hazard ratio for PARP expression and (A) overall survival and (B) progression free survival (PFS) in solid cancers according to chemotherapeutic treatment.

Result

…. Both direct and indirect methods (pooled HRs versus K-M curve data extraction) showed a correlation with poor OS (Direct: HR = 1.58, 95% CI = 1.33–1.88, P < 0.001; Indirect: HR = 1.41, 95% CI = 1.12–1.78, P < 0.001) (Supplementary Figure 2B).

Based on chemotherapy regimen, a significant poor OS was found with high PARP expression in breast cancer patients receiving neo-adjuvant chemotherapy of anthracycline and taxane (HR = 1.98, 95% CI = 1.04–3.78 P =0.04) and ovarian cancer patients receiving adjuvant chemotherapy of paclitaxel and carboplatin (HR = 1.15, 95% CI = 1.03–1.28, P =0.01), and platinum-based chemotherapy (agents not specified) (HR = 1.52, 95% CI = 1.15–2.02, P =0.003) (supplementary Figure 3A.)

According to antibody types, only Abcam (HR = 1.68, 95% CI = 1.19–2.36, P =0.003) and Santa Cruz Biotechnology …

Similarly, high PARP expression was associated with significantly worse PFS (HR = 1.05, 95% CI = 1.03–1.08, P < 0.001). PARP expression was associated with poor DFS in breast cancers and poor PFS in ovary cancers (Supplementary Figure 4). Subgroup analysis according to chemotherapy regimen, univariate analysis and multivariate analysis, ethnicity (Asian vs. Caucasian), and direct/indirect methods (pooled HRs vs. K-M curve data extraction) showed a significant relationship with poor outcome (Supplementary Figure 3B, 5and 6).

Discussion:

In our study, subgroup analysis according to chemotherapy regimen in breast and ovarian cancer patients showed poor OS and PFS in high PARP expression group regardless of the chemotherapy regimen. Although this alone cannot be a direct evidence of adverse biologic role of PARP expression because we could not include chemotherapy-free cohorts, the result blend smoothly into with the other findings from many other preclinical and clinical trial study results [54-57]. A previous preclinical study showed that treatment with PARP inhibitors leads to cell cycle arrest and cell death in cells with a homologous recombination deficiency (HRD) [58-60]. In pancreas, a clinical trial treating metastatic pancreatic cancer patients with germline BRCA1/2 mutations with olaparib extended the PFS [61]. In ovary, a clinical trial showed using olaparib, paclitaxel, and carboplatin increased the PFS in recurrent cancer patients [62]. All these lines of evidence indirectly support our results. To overcome aforementioned limitation, more evidence on the effect of treatment regimen is needed in the future.

Reviewer 2 Report

The systematic review presented by Prof. Yosef Chong and coworkers stablished different important conclusions based on a deep meta-analysis of actual and relevant published studies. The obtained conclusions are quite interesting and, what is more important, needed to finally understand the total action profile of PARP genes. Although, more of the conclusions are related to PARP-1, this fact does not rest significance to the study thus PARP-1 is the most studied and abundant PARP family gene. This review includes many variables such as size, organ, or even ethnicity, and a significant relation between higher PARP expression and overall survival was found. To date, several discrepancies between studies are shown because of different factors, sample size, organ type, etc. I want to highlight the ethnicity discussion section, since it is well referenced and explained. So that, I agree with the authors that a fully review had to be performed in order to clarify and gather together all the related research to PARP role in the development and prognosis of tumors. Because of this, I find that the current study is on a topic of relevance and general interest to the readers of the journal.

I found the paper to be overall well written and much of it to be well described, specially the “Material and methods” Section, although I found the Results Section kind of schematically written, I would be appreciated more fluid language. The design of the search strategy is a bit biased, thus other topics may be included, but it is totally understandable that it is difficult to establish a limit in this aspect. On the other hand, I found some of the description of the paper to be too detailed, while some other points seem to be written carelessly or rushed. Taking all in consideration, I recommend a minor revision before being considered to be published.

Minor comments:

  1. The manuscript is not well fitted in the journal template since the number of the pages is repeated and it is hard to know in what page you are. Fortunately, lines are numbered by the journal.
  2. The sentence of lines 306-309 should be rewritten since it is a bit confused, maybe the punctuation code is not correct or maybe the repeated used of the word “study” …
  3. The overall tone of the “Discussion” section should be reviewed since in some aspect, related to the other researchers and studies, is ruder that necessary.
  4. The reference format should be checked again in order to unify it. Some references include the DOI number, final page number or journal’s name is in short way and other do not.
  5. Relative to the supplementary Data, an index would be appreciated, and what is more important, at the end of the document, references are duplicated in other format. They should be deleted. Finally, in page 13, “Figure 8” should be replaced by “Supplementary Figure 8” just to follow the same format of the whole document.

Author Response

Reviewers 2 comments

The systematic review presented by Prof. Yosep Chong and coworkers stablished different important conclusions based on a deep meta-analysis of actual and relevant published studies. The obtained conclusions are quite interesting and, what is more important, needed to finally understand the total action profile of PARP genes. Although, more of the conclusions are related to PARP-1, this fact does not rest significance to the study thus PARP-1 is the most studied and abundant PARP family gene. This review includes many variables such as size, organ, or even ethnicity, and a significant relation between higher PARP expression and overall survival was found. To date, several discrepancies between studies are shown because of different factors, sample size, organ type, etc. I want to highlight the ethnicity discussion section, since it is well referenced and explained. So that, I agree with the authors that a fully review had to be performed in order to clarify and gather together all the related research to PARP role in the development and prognosis of tumors. Because of this, I find that the current study is on a topic of relevance and general interest to the readers of the journal.

I found the paper to be overall well written and much of it to be well described, specially the “Material and methods” Section, although I found the Results Section kind of schematically written, I would be appreciated more fluid language. The design of the search strategy is a bit biased, thus other topics may be included, but it is totally understandable that it is difficult to establish a limit in this aspect. On the other hand, I found some of the description of the paper to be too detailed, while some other points seem to be written carelessly or rushed. Taking all in consideration, I recommend a minor revision before being considered to be published.

Minor comments:

  1. The manuscript is not well fitted in the journal template since the number of the pages is repeated and it is hard to know in what page you are. Fortunately, lines are numbered by the journal.

Answer: We thank the reviewer for careful observation. We have changed the page number as per your suggestion.

     2. The sentence of lines 306-309 should be rewritten since it is a bit confused, maybe the punctuation code is not correct or maybe the repeated used of the word “study”

Answer: We thank the reviewer for improving our manuscript. We have modified the lines as below.

Previous version

In a previous meta-analysis on breast cancers by Qiao et al. the authors reported a sig-nificantly poor OS of high PARP expression in early breast cancer but not in locally advanced breast cancers. However, although many studies other than Aiad et al. study also include stage III or IV breast cancers (Table 1). Qiao et al did not include these studies in the locally advanced breast cancer group in the subgroup analysis but included them in the non-locally advanced breast cancer group, which is false Previous version

Modified version

In this study, we confirmed a significantly poor OS of high PARP expression in breast, ovary, lung and liver cancers. In a previous meta-analysis on breast cancers by Qiao et al., the authors reported a significantly poor OS of high PARP expression in early breast cancer but not in locally advanced breast cancers [42]. In contrast, through our systematic analysis, we found that many studies other than Aiad et al.’s included stage III or IV breast cancers (Table 1) [7]. However, Qiao et al. did not include these studies in the locally advanced breast cancer group in the subgroup analysis but included them in the non-locally advanced breast cancer group [42]. The results of Aiad et al.’s study should be interpreted with caution because it included only stage IIIB breast cancers, only 84 cases, that is relatively small, and only core needle biopsy samples (no other study used core needle biopsy samples), which could highly result in sampling bias [7]. In addition, Qiao et al. somehow combined two different results from one cohort of Aiad et al.’s study to perform a subgroup analysis of locally advanced breast cancer group [42]. The one result was targeting cytoplasmic immunoreactivity while another targeting nuclear immunoreactivity, which finally misleads to the wrong conclusion that high PARP expression is associated with better survival in locally advanced breast cancers and poor survival in early breast cancers. In this study, other than that, we included all results from additional studies and found a higher statistical significance of high PARP expression to the poor OS in overall breast cancers. Similarly, the same relationship between PARP expression and poor OS was found in ovary, lung, and liver cancers.

    3. The overall tone of the “Discussion” section should be reviewed since in some aspect, related to the other researchers and studies, is ruder that necessary.

Answer: Thank you for your valuable suggestion. We have deleted those offensive expression.

    4. The reference format should be checked again in order to unify it. Some references include the DOI number, final page number or journal’s name is in short way and other do not.

Answer: We are thankful for careful observation. We have modified the references according to your suggestion.

   5. Relative to the supplementary Data, an index would be appreciated, and what is more important, at the end of the document, references are duplicated in other format. They should be deleted. Finally, in page 13, “Figure 8” should be replaced by “Supplementary Figure 8” just to follow the same format of the whole document.

Answer: We thank the reviewer for careful observation. We have made the changes as per your suggestion as below.

  1. We have made an index in the first pages of the supplementary file as below.
  2. We have removed the duplicated reference file.
  3. We have replaced the “Figure 8” by “Supplementary Figure 9”, because of one more additional figure.

Reviewer 3 Report

Thakur and colleagues have written an extensive review regarding the expression levels of Poly-ADP-ribose polymerase and the relationship with overall survival in solid cancers.

The review is well written and detailed therefore as such represents a sound resource for both the PARP and cancer communities. The search strategy is well planned and robust. The quality controls and filtering of the data are also in line with standard procedures. The authors also take time to discuss well the limitations of the study and propose additional future work which could address these concerns. 

One comment which concerns me is that they state that the finding that PARP1 expression is and poor overall survival in ovary, lung and liver cancers is a novel finding from this study but the authors already mentioned the research below in their introduction which makes me wonder as to the exact novelty of the data presented here. A more detailed explanation of the limitations of the previous studies, may help to increase the significance or impact of the findings presented here. 

https://www.frontiersin.org/articles/10.3389/fonc.2020.00958/full#h4

https://link.springer.com/article/10.1007/s12253-020-00856-6#Sec6

Author Response

Reviewers 3 comments

Thakur and colleagues have written an extensive review regarding the expression levels of Poly-ADP-ribose polymerase and the relationship with overall survival in solid cancers.

The review is well written and detailed therefore as such represents a sound resource for both the PARP and cancer communities. The search strategy is well planned and robust. The quality controls and filtering of the data are also in line with standard procedures. The authors also take time to discuss well the limitations of the study and propose additional future work which could address these concerns. 

One comment which concerns me is that they state that the finding that PARP1 expression is and poor overall survival in ovary, lung and liver cancers is a novel finding from this study but the authors already mentioned the research below in their introduction which makes me wonder as to the exact novelty of the data presented here. A more detailed explanation of the limitations of the previous studies, may help to increase the significance or impact of the findings presented here. 

https://www.frontiersin.org/articles/10.3389/fonc.2020.00958/full#h4

https://link.springer.com/article/10.1007/s12253-020-00856-6#Sec6

Answer: Thank you for your comment. Our study is meaningful because there was no meta-analysis on ovary, lungs and liver according to PARP expression and survival. The first link you mentioned earlier was efficacy of PARP1 inhibitors clinically according to OS, PFS in ovary. Of course this is really important study to mention so we included it and explained about it in our discussion. The only meta-analysis study about the prognostic significance of PARP1 expression was in breast and it showed a few flaws in the subgroup analysis. We carefully mentioned this in our manuscript to highlight the novelty of this study.

Round 2

Reviewer 1 Report

No additional comment